# Genome Organizations and Functional Analyses of a Novel Gammapartitivirus from *Rhizoctonia solani* AG-1 IA Strain D122

**DOI:** 10.3390/v13112254

**Published:** 2021-11-10

**Authors:** Meiling Zhang, Zhenrui He, Xiaotong Huang, Canwei Shu, Erxun Zhou

**Affiliations:** 1Guangdong Province Key Laboratory of Microbial Signals and Disease Control, College of Plant Protection, South China Agricultural University, Guangzhou 510642, China; meilingzhangsy@163.com (M.Z.); zhenruihe@163.com (Z.H.); huangxiaotong98@163.com (X.H.); 2School of Biological and Food Engineering, Anyang Institute of Technology, Anyang 455000, China

**Keywords:** mycovirus, *Gammapartitivirus*, hypovirulence, *Rhizoctonia solani*, transcriptome, biocontrol

## Abstract

Here, we describe a novel double-stranded (ds) RNA mycovirus designated Rhizoctonia solani dsRNA virus 5 (RsRV5) from strain D122 of *R**hizoctonia solani* AG-1 IA, the causal agent of rice sheath blight. The RsRV5 genome consists of two segments of dsRNA (dsRNA-1, 1894 bp and dsRNA-2, 1755 bp), each possessing a single open reading frame (ORF). Sequence alignments and phylogenetic analyses showed that RsRV5 is a new member of the genus *Gammapartitivirus* in the family *Partitiviridae*. Transmission electron microscope (TEM) images revealed that RsRV5 has isometric viral particles with a diameter of approximately 20 nm. The mycovirus RsRV5 was successfully removed from strain D122 by using the protoplast regeneration technique, thus resulting in derivative isogenic RsRV5-cured strain D122-P being obtained. RsRV5-cured strain D122-P possessed the traits of accelerated mycelial growth rate, increased sclerotia production and enhanced pathogenicity to rice leaves compared with wild type RsRV5-infection strain D122. Transcriptome analysis showed that three genes were differentially expressed between two isogenic strains, D122 and D122-P. These findings provided new insights into the molecular mechanism of the interaction between RsRV5 and its host, D122 of *R. solani* AG-1 IA.

## 1. Introduction

Mycoviruses, also known as fungal viruses, are ubiquitous in almost all major groups of filamentous fungi, oomycetes and yeasts [1,2,3,4]. Most mycoviruses possess either double-stranded RNA (dsRNA) genomes or positive sense (+) single-stranded RNA (ssRNA) genomes, and some have negative-sense (−) ssRNA or single-stranded DNA (ssDNA) genomes [5,6]. Although many mycoviruses do not cause any visible abnormal symptoms in their fungal and oomycete hosts, a few can reduce their hosts’ pathogenicity. This phenomenon is known as hypovirulence [2,7]. Mycovirus-associated hypovirulent traits of plant pathogenic fungi are important biocontrol resources for the control of plant fungal diseases, e.g., the +ssRNA mycovirus Cryphonectria hypovirus 1 (CHV1) against chestnut blight caused by *Cryphonectria parasitica* [7], and the control of rapeseed stem rot caused by *Sclerotinia sclerotiorum* with Sclerotinia sclerotiorum hypovirulence-associated DNA virus 1 (SsHADV-1) [8]. Therefore, scientists are attempting to discover more mycoviruses, so as to deepen our understanding of their diversity, evolution and biocontrol potential in the biocontrol of plant fungal diseases.

With the widespread application of molecular techniques and sequencing technologies in recent years, our knowledge of partitivirus diversity has increased markedly. In recent years, more and more partitiviruses have been identified in fungal, plant and protozoa samples. Mycoviruses in the family *Partitiviridae* commonly contain bisegmented dsRNA genomes, which are 1300–2500 bp in length and encompass one large open reading frame (ORF) per segment [1,9]. The segment encoding the RNA-dependent RNA polymerase (RdRp) protein is named dsRNA-1, and that encoding the coat protein (CP) is designated as dsRNA-2 [9]. Currently, there are five approved genera in this family: *Alphapartitivirus*, *Betapartitivirus*, *Cryspovirus*, *Deltapartitivirus* and *Gammapartitivirus* [10]. It is worth mentioning that outside the approved genera, epsilonpartitivirus and zetapartitivirus genera are also proposed, and a large number of unclassified viruses have been discovered [11] The transmission difficulty of mycoviruses is a key factor constraining their adaptation to the environment [12,13]. In fungal hosts, partitivirus transmits horizontally via hyphal anastomosis and vertically via gamete and spore formation [9,14]. In plant hosts, dispersal from host individuals is then by pollen or seeds [15].

The notorious soil-borne fungus *Rhizoctonia solani* Kühn [teleomorph: *Thanatephorus cucumeris* (Frank) Donk], a collective species, is a devastating fungal plant pathogen that can infect many vital crops such as rice, maize and peanut worldwide [3,16,17,18]. Earlier studies have detected various sized dsRNA segments in natural populations of *R**. solani* from AG-1 to AG-13 [4]. Previously, our laboratory isolated and characterized five dsRNA mycoviruses from the *R. solani* AG-1 IA. Two of these viruses were isolated from strains GD-11 and A105 of *R. solani* AG-1 IA and belonged to the genus *Alphapartitivirus* in the family *Partitiviridae*. The nucleotide sequence and genomic organization of the three novel mycoviruses, Rhizoctonia solani endornavirus 1 (RsEV1), Rhizoctonia solani dsRNA virus 1 (RsRV1) and Rhizoctonia solani partitivirus 5 (RsPV5), were also reported by our laboratory [16,17,18,19,20]. Most fungal viruses that infect their hosts are symptomless, which makes the exploration of novel viruses a great challenge; thus, a large number of mycoviruses in *R. solani* remain undiscovered.

In this study, we isolated a dsRNA virus named Rhizoctonia solani dsRNA virus 5 (RsRV5) from strain D122 of *R. solani* AG-1 IA with a unique genome organization. We also studied the viral molecular characterization, phylogenetic analyses and particle morphology. In addition, the mycovirus RsRV5 was eliminated via the protoplast regeneration technique, and its impacts on the biological characteristics of *R. solani* were also investigated. Finally, transcriptome technology was used to explore the molecular mechanism of the interaction between RsRV5 and *R. solani* AG-1 IA.

## 2. Materials and Methods

### 2.1. Fungal Strains and Cultural Conditions

The *R. solani* AG-1 IA strain D122 was originally isolated from a typical rice sheath blight disease lesion collected from a rice field in Guangdong province, China. *R. solani* AG-1 IA strain GD118, a virulent virus-free strain, served as a virus recipient in the viral transmission experiment and is maintained in our laboratory [21]. All *R. solani* strains were cultured on potato dextrose agar (PDA) medium at 28–30 °C and were stored on PDA slants at 4 °C.

### 2.2. Extraction and Detection of dsRNA

To extract the dsRNA of mycovirus, the mycelia of strain D122 were cultured on a cellophane membrane overlaying a PDA plate for 6 days. Mycelia were collected and ground with a mortar and pestle into fine powder in the presence of liquid nitrogen. Subsequently, the nucleic acids of mycovirus were extracted from 15 g of frozen mycelia by selective absorption to the columns of cellulose powder CF-11 (SIGMA-ALDRICH, Inc., Louis, MO, USA) according to the method described by Morris and Dodds [22], with minor modifications. The nucleic acid of mycovirus was purified by treating with DNase I and S1 nuclease (TaKaRa, Dalian, China) to eliminate contaminating DNA and ssRNA, respectively. The quality and concentration of purified dsRNAs were detected by using electrophoresis on 1.0% agarose gels and stored at −80 °C.

### 2.3. cDNA Cloning, Sequencing and Sequence Analysis

The dsRNAs extracted from *R. solani* AG-1 IA strain D122 were purified and used as templates for complementary DNAs (cDNAs) cloning. Cloning the cDNAs for the dsRNAs from strain D122 by random primer-mediated PCR, sequencing them and analyzing the sequences were carried out using the procedures described by Zheng et al., with minor modifications [18,20]. To clone the terminal sequences of each dsRNA, the 5′ and 3′ end cDNA amplifications were performed using a slightly modified rapid amplification protocol of cDNA ends, as described by Darissa et al. [23].

Searches for ORFs in each full-length cDNA sequence were conducted using the ORF Finder Program of the National Center for Biotechnology Information (NCBI). Sequence analysis of conserved domains was performed using the NCBI BLASTp program. A phylogenetic tree was constructed using the neighbor-joining (NJ) in MEGA6.0 and a bootstrap test was estimated by 1000 replications [24].

### 2.4. Northern Hybridization

Northern hybridization was performed to verify the authenticity of the cDNA sequences generated from mycovirus in strain D122 of *R. solani* AG-1 IA. We mixed 8 µg of dsRNA from strain D122 with formaldehyde load dye (1:2) and incubated for 10 min at 65 °C to denature, and immediately placed on ice for 1 min. The denatured RNA was separated by 1% agarose gel with 5 *v*/cm electrophoresis for 2 h. It was then transferred from the agarose gel to an Immobilon^™^-Ny^+^ membrace (Millipore, Bedford, MA, USA) in 20 × SSC buffer for 16–24 h, and UV was used to cross-link RNA to nylon membranes. The cDNA probe 1 corresponding to the dsRNA-1 sequence is 373 bp in length and was obtained using specific primers (RdRpProbeF: 5′-GGATGAAGTCAAGCAG-3′ and RdRpProbeR: 5′-GGCGACAGTACGATGG-3′). The cDNA probe 2 corresponding to the dsRNA-2 sequence is 473 bp in length and was obtained using specific primers (CPProbeF: 5′-CGAACGCAACAGAACA-3′ and RdRpProbeR: 5′-GAAACACCCGAAAAGTCA-3′). These probes were labeled with the DIG High-Prime DNA labeling and detection starter kit I (Roche Diagnostics GmbH, Roche Applied Science, Mannheim, Germany), respectively. Then, we put the membrane in a glass tube with 15 mL of DIG Easy Hyb buffer and incubated the glass tube at 68 °C for 30 min. The probes were placed in a boiling water bath for 5 min and then cooled rapidly on ice, and the denatured probes were added directly to the hybridization solution. Hybridization was carried out under high stringency conditions in a hybridization solution for 8 h at 68 °C. Post-hybridization washes were conducted twice with primary (2 × SSC, 0.1% SDS) and secondary (0.1 × SSC, 0.1% SDS) wash buffer. Hybridization signals were detected by chemiluminescence using a CDP-Star Detection kit (GE Healthcare, Life Sciences, Bristol, UK).

### 2.5. Purification of Viral Particles and Electron Microscopy

We purified the viral particles using the sucrose-gradient method described by Sanderlin and Ghabrial [25] with minor modifications. Transmission electron microscopy (TEM) (Tecnai 12, Amsterdam, Netherlands) was used to observe viral particles stained with 2% (*w*/*v*) phosphotungstate solution (pH 7.4). The nucleic acid from viral particles was extracted with phenol, chloroform and isoamyl alcohol, and separated by electrophoresis in 1% (*w*/*v*) agarose gel [18].

### 2.6. Virus-Elimination

To eliminate mycovirus from strain D122, hyphal tipping, ribavirin treatment, protoplast regeneration and a combination of the two approaches were conducted. Transfection was performed via the inoculation of protoplasts of strain GD118 with purified viral particles according to the method of Sasaki et al. and Hillman et al. [26,27]. The protoplasts of virus-free strains GD-118 and D122 were prepared using the method described previously by our laboratory [21]. Protoplasts were regenerated in a regeneration medium (yeast extract 1.0 g/L; enzymatic casein hydrolysate 1.0 g/L; glucose 0.5 M; agar 15 g/L) at 26 °C for 2–3 days. Mycelial plugs were cut at random from the regenerated colonies and transferred to fresh PDA plates. For ribavirin treatment, mycelial plugs of strain D122 were inoculated on water agar (WA) medium containing ribavirin (100 µM) and cultured at 28–30 °C. For hyphal tipping, we cultured RsRV5-infection strain D122 on WA, using a surgical knife to excise mycelial plugs containing the tip of a single hyphal under the microscope, and each small piece was cultured on WA. The presence or absence of RsRV5 was confirmed by extracting dsRNA and RT-PCR using specific primers. The new mycovirus-cured strain was designated as D122-P.

### 2.7. RNA-Seq and Data Analysis

The virus-infected strain D122 and its isogenic virus-cured strain D122-P were cultured on PDA plates covered with cellophane membrane in an incubator at 28–30 °C for 5 days. Fungal mycelia were collected for the isolation of total RNA using Trizol (TaKaRa, Dalian, China). For RNA-Seq library preparation, 5 µL of total RNA per sample was extracted. The RNA quality and integrity were determined by Agilent 2100 Bioanalyzer (Agilent 2100) and 1% agarose gel electrophoresis, respectively. Illumina complementary DNA libraries were generated using NEBNext^®^Ultra™II RNA Library Prep Kit for Illumina^®^ according to the manufacturer’s instructions. The quality check of the library was performed by the Agilent 2100 Bioanalyzer and ABI Real-Time PCR System. RNA deep sequencing was carried out by the Illumina HiSeq 2000 (Illumina, San Diego, CA, USA). RNA-seq was performed for two strains with 2 replicates per each. Initial quality control of the sequencing data was performed using CASAVA (V1.8.2) software. The unqualified reads were filtered out and contained paired-end reads shorter than 75 bp, as well as low-quality scores (<20) in the raw data and the adapter sequence. The clean reads of *Rhizoctonia solani* AG-1 IA strain D122 and D122-P were mapped to the reference sequence of *R. solani* AG-1 IA using Hisat (version 0.1.6). Gene expression level was calculated using rsem (V1.2.6) software. The differential expression of genes (DEGs) was assessed using the EdgeR (V3.4.2) program, along with fold change (FC) >2 and false discovery rate (FDR) <0.05 as a screening criterion. The FDR was obtained by correcting the *p*-value of gene differences. Then, categorization and expression statistics of Cufflinks predicted transcripts with variable splicing events using ASprofile (V1.0.4) software.

## 3. Results

### 3.1. Detection of dsRNAs in R. solani Strains

We observed that strain D122 produced far fewer sclerotia on PDA compared to a typical strain GD118 of *R. solani* AG-1 IA. Based on previously reported phenotypes of fungal strains infected with various mycoviruses, we presumed that strain D122 may be infected by one or more mycoviruses [1,4]. When screening for dsRNAs from the strain D122 using the CF-11 cellulose chromatography method, two dsRNA segments (dsRNA-1 and dsRNA-2) of about 2 kb were observed in the strain D122 (Figure 1). Subsequently, S1 nuclease (active against ssDNA or ssRNA) and DNase I (active against ssDNA and dsDNA) were used to confirm the properties of the extracted viral nucleic acids. The results showed that the viral nucleic acids could not be digested by both S1 nuclease and DNase I (Figure 1), suggesting that strain D122 was infected by dsRNA mycoviruses.

### 3.2. Nucleotide Sequence and Amino Acid Sequence of RsRV5

The complete nucleotide sequences of the two dsRNA segments, designated as dsRNA-1 and dsRNA-2, were determined from a series of cDNA clones spanning the entire length of each dsRNA segment. According to a report by Nibert et al. [9], the segment encoding the RNA-dependent RNA polymerase (RdRp) protein was designated as dsRNA-1 (the longer segment in almost all strains), and the one encoding the capsid/coat protein (CP) was designated as dsRNA-2. The sequences of these dsRNA segments were deposited in GenBank with accession numbers MG597242 and MG597241. The genetic organization of the two dsRNAs is shown in Figure 2. The complete nucleotide sequences of two dsRNA segments of Rhizoctonia solani dsRNA virus 5 (RsRV5) were 1894 bp (dsRNA-1) and 1755 bp (dsRNA-2) in length, similar to the size estimated by agarose gel electrophoresis. The bisegmented genome of RsRV5 was confirmed by Northern blotting (Figure 2c). The 5′ untranslated region (UTR) and 3′-UTR of the mycovirus were 29 bp and 107 bp long in dsRNA-1 and 132 bp and 132 bp long in dsRNA-2 (Figure 2a). The 5′-terminal sequences of both dsRNAs have a high similarity, which might be involved in the replication cycle of the dsRNAs [18,19]. This indicates that the 5′-termini of the coding strands of the two dsRNAs contains conserved sequences. Notably, adenine-rich regions were detected in the 3′-UTRs of the two dsRNAs (Figure 2b), as shown in other members of *Partitiviridae* [28,29,30], which were similar to interrupted poly (A) tails.

Analysis of the determined sequence revealed that dsRNA-1 contains a single open reading frame (ORF) starting at nt 30 and ending at nt 1787 on its plus strand. The single ORF1 encodes a 585-aa protein with a predicted molecular mass of 67.36 kDa. A sequence search with BLASTP suggested that this protein was most closely related to the RdRp of some viruses in the family *Partitiviridae*, such as the Penicillium aurantiogriseum partiti-like virus [31]. The sequence of dsRNA-2 contains a single ORF, ORF2, starting at nt 133 and ending at nt 1623, which encodes a 496-aa protein with a molecular mass of 54.98 kDa. A sequence search with BLASTP suggested that this protein was most closely related to the coating protein (CP) of the family *Partitiviridae*.

### 3.3. Phylogenetic Analysis of the Partitivirus RsRV5

To analyze the relationship between RsRV5 and other dsRNA mycoviruses, a phylogenetic tree (Figure 3) based on the RdRp sequences of RsRV5 and 29 other selected viruses in the families *Partitiviridae* and *Totiviridae* was constructed using the neighbour-joining method [24].

The result of the phylogenetic analysis showed that RsRV5, the Hubei tetragnatha maxillosa virus 8 [32], the Wuhan fly virus 5 [32], the Beihai barnacle virus 12 [32] and the Penicillium aurantiogriseum partiti-like virus [31] were clustered together in a distinct group belonging to the genus *Gammapartitivirus* with sequence identities of 32.69%, 30.20%, 32.62% and 48.02%, respectively. Therefore, genome organizations, amino acid sequence alignments and phylogenetic analyses all support that RsRV5 is a new member of the genus *Gammapartitivirus* within the family *Partitiviridae*.

### 3.4. Viral Particles

The purified viral particles were successfully obtained from the mycelia of strain D122, and uniform viral particles were observed under TEM. TEM observation revealed that the virus particles purified from strain D122 were isometric with an approximate diameter of 20 nm (Figure 4a), comparable to the Rhizoctonia solani dsRNA virus 3 (RsRV3) previously reported in our laboratory [19]. Nucleic acids extracted from the viral particles appeared as two bands at ~2 kb in 1% agarose gel electrophoresis. The size and brightness of the two dsRNA segments released from the purified viral particles were similar to those of dsRNA 1 and dsRNA 2 extracted directly from the mycelia of strain D122 (Figure 4b). This suggests that the segment numbers of the dsRNAs extracted from viral particles are the same as those dsRNAs extracted directly from mycelia, which is akin to the replication principle of the partitivirus [9]. These results also demonstrated that both the viral particles and dsRNAs extracted from the mycelia of strain D122 belong to the same mycovirus, RsRV5. Viral proteins purified from strain D122 were subjected to SDS-PAGE analysis. The results showed the presence of two major structural proteins with a molecular mass of about 60 kDa (Figure 4c). The size of the isolated proteins is similar to that predicted for RdRp and CP based on dsRNA sequence analysis, respectively. Thus, two of the proteins are assumed to be the RsRV5 structural proteins.

### 3.5. The Mycovirus Affects the Fungal Host Phenotypes in Strain D122

To investigate whether the mycovirus was responsible for this impaired growth, we first attempted to eliminate it from the fungal host by hyphal tipping and ribavirin treatment. Nevertheless, repeated attempts were unsuccessful. Therefore, protoplast regeneration was adopted to continue the virus elimination experiment. We obtained the candidate protoplast regeneration strain D122-P. The presence or absence of RsRV5 was confirmed by electrophoresis of the genomic dsRNAs (Figure 5a) and by RT-PCR, using specific primers for the RsRV5-dsRNA-1 and RsRV5-dsRNA-2 (Figure 5b). The specific dsRNA segment was detected in the original mycovirus-infected strain, D122, but not in the protoplast regeneration-derived isogenic strain D122-P (Figure 5a). Successful elimination was also confirmed by RT-PCR analysis (Figure 5b). These results indicated that the RsRV5 originated in strain D122 was successfully eliminated in the protoplast regeneration-derived isogenic strain D122-P.

Colony morphologies of these two isogenic strains D122 and D122-P, grown under the same conditions, were compared (Figure 6a). The results indicated that D122-P had a different phenotype, including a faster growth rate (Figure 6b), more sclerotia and dark pigmentation on the PDA plate when compared with D122 (Figure 6a). On the contrary, the RsRV5-infection strain D122 had an abnormal phenotype. In addition, the effect of RsRV5 on fungal virulence was evaluated based on lesion sizes on rice leaves caused by the two isogenic strains D122 and D122-P. Pathological tests showed that the average lesion areas caused by D122 were smaller than those caused by D122-P (Figure 6c), suggesting that RsPV5 induced hypovirulence in the virus-infected strain D122.

### 3.6. RNA-seq Analysis of Rhizoctonia solani AG-1 IA Response to RsRV5 Infection

To identify genes of *Rhizoctonia solani* AG-1 IA that play key roles in response to RsRV5 infection, RNA-seq technology was applied to compare the expression of fungal host genes in isogenic strains D122 and D122-P. Data analysis showed that for samples of strains D122 and D122-P, there were a total of 33 million and 31 million reads, respectively, of which an average of 73.88% and 76.17% reads, respectively, were aligned to the *Rhizoctonia solani* AG-1 IA. In this study, we used absolute logFC > 1 and FDR < 0.05 to define DEGs. Compared to the gene expression data of RsRV5-infection strain D122, a total of three genes (AG1IA_06216, AG1IA_06615 and AG1IA_09435) as candidates in which expression was altered were found in strain D122-P, with two up-regulated (AG1IA_06216 and AG1IA_06615) and one down-regulated (AG1IA_09435). Gene AG1IA_09435 was supposed to encode a sulfotransferase family domain-containing protein. Gene AG1IA_06216 and gene AG1IA_06615 were predicted to encode a hypothetical protein, respectively. The “volcano plot” and “MA plot” showed the distribution of the DEGs between strains D122 and D122-P (Figure 7). A correlation check of RNA-seq revealed similar expression patterns and R^2^ > 0.8 among the two biological repetitions of the *R. solani* AG-1 IA strains D122 (D122a and D122A) and D122-P (D122-Pa and D122-PA), indicating that the RNA-seq data were reliable (Figure 8).

Subsequently, computational analysis was performed to determine the type and number of alternative splicing (AS) events in strains D122 and D122-P. The results showed that five common types of AS events could be examined, including exon skipping, intron retention, alternative 5′ss splice or alternative 3′ss splice, alternative transcription start site (TSS) and alternative transcription terminal site (TTS) (Table 1). The alternative 5′ss splice or alternative 3′ss splice in strains D122 and D122-P is the most proportional form of AS, followed by the alternative transcription start site. Meanwhile, we found that 33,089 *R. solani* AG-1 IA genes underwent AS, with 65,535 AS events in this study. These RNA-seq results suggest the complexity of the interaction mechanisms between mycovirus and its host fungus and pave the way for further research of mycovirus-fungal host interactions.

## 4. Discussion

In recent years, more and more mycoviruses have been discovered, with some being hypovirulence-associated mycoviruses with potential utilization value in the biological control of plant fungal diseases [33]. Investigating the rich diversity of mycoviruses has provided more information on virus diversity, evolution and dealing with emerging viral diseases affecting plants, animals and humans [1,4]. In this study, we described the complete nucleotide sequence, genome organization and biological characterization of a novel mycovirus RsRV5 which infected the plant pathogenic fungus *R. solani* AG-1 IA, as well as the interactions between fungal hosts and their mycovirus. This research expands our knowledge of the diversity and evolution of *R. solani* mycoviruses. According to Nibert et al. [9], partitiviruses possess two essential dsRNA genome segments, dsRNA-1 and dsRNA-2, each 1300~2500 bp in length and containing one long ORF on one of the RNA strands, i.e., the plus strand. Previous reports have shown that the genomes of both Rhizoctonia solani partitivirus 3 (RsPV3) and Rhizoctonia solani partitivirus 4 (RsPV4) of the family *Partitiviridae* contain two separate dsRNA segments, each with a single ORF. ORF1 from RsPV3 and RsPV4 encodes a putative RdRp, while ORF2 from RsPV3 and RsPV4 encodes a putative CP [34]. Similar to viruses of the family *Partitiviridae*, the genomic sequence of RsRV5 also consists of two essential dsRNA segments, i.e., dsRNA-1, 1894 bp and dsRNA-2, 1755 bp. The 5′-terminal sequences of two dsRNAs are highly conserved, which is a unique structure to the family Partitiviridae. Previous studies indicate that the terminal structure is often involved in the transcription, replication and packaging of viral RNA. It has been reported that the 5′-terminal sequence of dsRNA-1 and dsRNA-2 of the R. solani 717 virus contains inverted repeats capable of forming stem-loop structures and is highly conserved [30]. The ‘interrupted’ poly-A tails are detected in the 3′-UTRs of the two dsRNAs, which have also been reported in other members of Partitiviridae. The Pleurotus ostreatus virus 1(PoV1), a partitivirus, has been reported to possess ‘interrupted’ poly-A tails [28]. It is worth noting that the poly-A tails of some other viruses are involved in the virus replication. For example, the Saccharomyces cerevisiae L-BC RNA replicase recognizes the 3′-UTR sequence of the Saccharomyces cerevisiae L-A virus [35]. The role of the ‘interrupted’ poly-A tails of RsRV5 in viral replication deserves further exploration.

Moreover, the phylogenetic analysis with RdRp sequences placed RsRV5 in a distinctive clade with members of *Gammapartitivirus* in the family *Partitiviridae* (Figure 3). Proteins encoded by dsRNA-1 shared the highest aa sequence identities to the RdRp of members of the family *Partitiviridae*. TEM analysis of viral particles revealed the RsRV5 virions were isometric particles ~20 nm in diameter. Its size is similar to that of the families *Partitiviridae* (25 to 42 nm), *Chrysoviridae* (30 to 40 nm) and *Totiviridae* (30 to 40 nm). It is not clear whether the two dsRNAs of RsRV5 are encapsulated in separate virus particles or within the same ones. According to Nibert et al. [9], the dsRNA segments of the genomes of the mycoviruses in *Partitiviridae* are individually encapsulated. Therefore, we thoroughly explored the viral particles, as well as the genome segment characteristics of RsRV5. These results indicated that RsRV5 is evolutionarily related to members of the family *Partitiviridae*. In conclusion, based on the dsRNA number, viral particles, genomic organization and phylogenetic analysis, RsRV5 was identified as a member belonging to the genus *Gammapartitivirus* in the family *Partitiviridae*, according to the demarcation criteria for these mycoviruses. An attempt at the horizontal transmission of hypovirulence and RsRV5 from D122 to the virulent strain GD118 of *R. solani* AG-1 IA by the dual culture technique was unsuccessful in this study. We also failed to isolate virus-free strains via hyphal tipping and ribavirin treatment, suggesting that RsRV5 is maintained stably and systemically in this natural fungal host. In contrast, our previous report demonstrated that an endornavirus RsEV1 could be successfully transmitted from strain GD-2 to strain GD-118P of *R. solani* AG-1 IA [16]. The reason for the unsuccessful transfection of RsRV5 in this study needs further investigation. Partitiviruses were originally thought to cause cryptic infections and be persistent with the host fungus [33]. However, we attempted to obtain a virus-free strain by protoplast regeneration, which was successful. It is worth mentioning that RsRV5-cured strains D122-P derived from protoplast regeneration showed higher virulence on rice leaves, including both more lesion numbers and larger lesion size, compared with the isogenic original RsRV5-infected strain D122 of *R. solani* AG-1 IA. These results suggest that RsRV5 confers hypovirulence in strain D122, which has the potential to be used as a biological control agent against rice sheath blight caused by *R. solani* AG-1 IA. So far, the biological control of chestnut blight using Cryphonectria hypovirus 1 (CHV1) is the most successful example [36]. Thus, the use of RsRV5 as a biological control agent for rice sheath blight still needs to overcome great difficulties. It has been previously reported that a large number of partitiviruses have been identified in phytopathogenic fungi, but few of them have been involved in the hypovirulence of their host fungi [4]. Among the approximately 100 mycoviruses that have been found in *R. solani* isolates, several have been associated with hypovirulence. For example, the RSPV2 isolated from strain GD-11 of *R. solani* AG-1 IA showed reduced mycelial growth and hypovirulence to rice leaves [18]. A novel Gammapartitivirus infecting Colletotrichum is reported to be associated with conidial production in fungal hosts [37]. In addition to partitiviruses, the RsEV1 in the family *Endornaviridae* can also cause metabolic disorders in the host and lead to reduced virulence [17]. In this study, RNA-seq based transcriptomic technology was used to explore the interactions between mycovirus and host fungus. We tried to explore the effect of the mycovirus infection on the gene expression of the host fungus by comparing the transcriptional differences between two isogenic strains, i.e., RsRV5-infected strain D122 and RsRV5-cured strain D122-P. DEG analysis showed that two genes were upregulated, and one was downregulated after RsRV5 infection. These DEGs might be the key factors responsible for the abnormal phenotype of the fungal host. However, the reason why there are so few DEGs after a viral infection has been a persistent question. In contrast, according to Qu et al. [38], by comparing the gene expression of hypovirulent Sclerotinia sclerotiorum strain DT-8 and virulent virus-free strain DT-8VF, a total of 3110 statistically significant DEGs were found in strain DT-8, with 1741 up-regulated and 1369 down-regulated. Previous reports demonstrated that the AS is an important mechanism of gene regulation and plays an essential role in proteomic diversity and functional complexity [39]. In order to determine the effect of the mycovirus RsRV5 on the AS of *R. solani* AG-1 IA, we tried to explore what types and amounts of AS events occur at the overall level during a period of infection. A total of 33,089 *R. solani* AG-1 IA genes with AS variants were detected through the transcriptomic technology, suggesting that AS might have vital, multiple biological roles in the fungus. In conclusion, we will focus future studies on the battle between RsRV5 and *R. solani* AG-1 IA and thoroughly investigate the hypovirulent characteristics, which might provide new insights into the control of rice sheath blight.

## Figures and Tables

**Figure 1 viruses-13-02254-f001:**
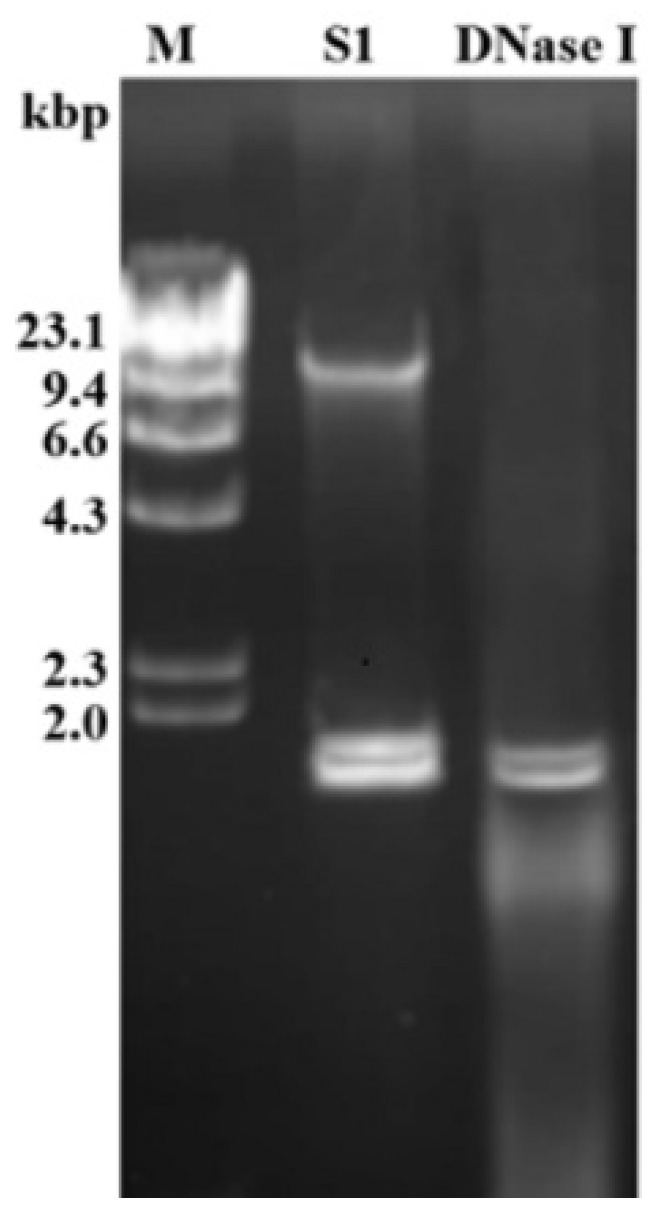
Electrophoresis analysis of enzyme-treated nucleic acid samples on 1% agarose gel. The nucleic acid samples were treated with S1 nuclease and DNase I, respectively. M: molecular markers (λ DNA digested with *Hind* III); Kbp: kilobase pair; gDNA: genomic DNA of the fungal strain D122.

**Figure 2 viruses-13-02254-f002:**
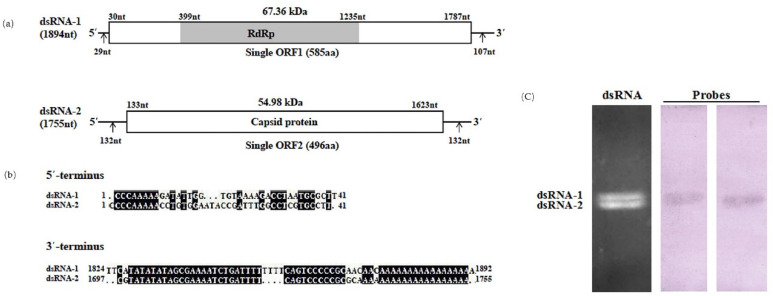
Molecular characteristics of RsRV5. (**a**) Schematic diagram of the genomic organization of RsRV5; (**b**) Multiple alignment of the terminal regions of RsRV5 genome; (**c**) Northern blot detection of dsRNA-1 and dsRNA-2 using digoxigenin-labeled probes 1 and 2.

**Figure 3 viruses-13-02254-f003:**
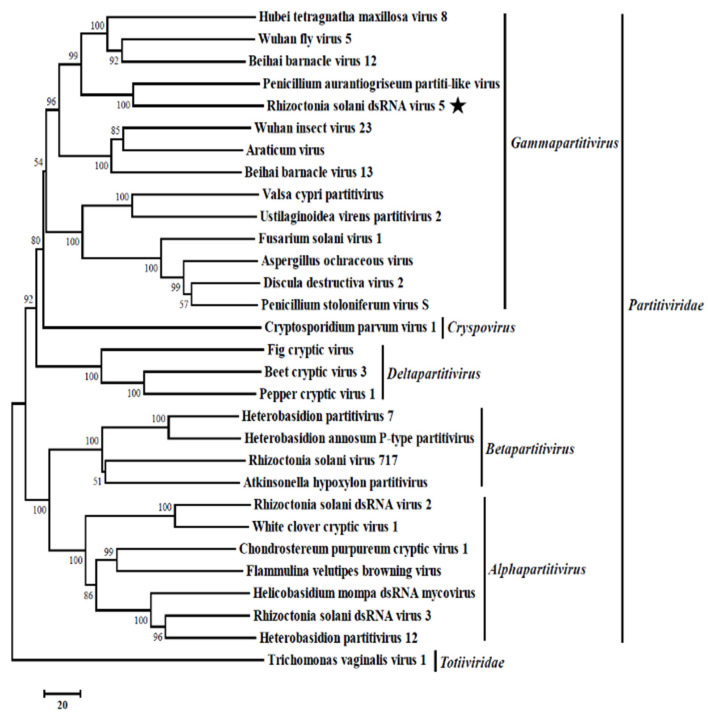
Phylogenetic relationship of RdRp genes of Rhizoctonia solani dsRNA virus 5 (RsRV5) and selected dsRNA viruses. An unrooted phylogenetic tree was constructed by a neighbor-joining method based on multiple amino acid sequence alignments of the RdRp using MEGA6.0. Bootstrap values (%) obtained with 1000 replicates are indicated on branches. Rhizoctonia solani dsRNA virus 5 (indicated with black star) was grouped into the genus *Gammapartitivirus*. Relate viruses and GenBank accession numbers are as follows: Hubei tetragnatha maxillosa virus 8 (YP_009337885.1), Wuhan fly virus 5 (YP_009342458.1), Beihai barnacle virus 12 (YP_009333370.1), Penicillium aurantiogriseum partiti-like virus (YP_009182157.1), Wuhan insect virus 23 (APG78216.1), Araticum virus (ASV45859.1), Beihai barnacle virus 13 (YP_009329869.1), Valsa cypri partitivirus (AIS37548.1), Ustilaginoidea virens partitivirus (AHH35116.1), Fusarium solani virus 1 (NP_624350.1), Aspergillus ochraceous virus (ABC86749.1), Discula destructiva virus 2 (NP_620301.1), Penicillium stoloniferum virus S (AAN86834.2), Cryptosporidium parvum virus 1 (AAC47805.1), Fig cryptic virus (CBW77436.1), Beet cryptic virus 3 (AAB27624.1), Pepper cryptic virus 1 (AEJ07890.1), Heterobasidion partitivirus 7 (AHA82537.1), Heterobasidion annosum P-type partitivirus (AAL79540.1), Rhizoctonia solani virus 717 (NP_620659.1), Atkinsonella hypoxylon partitivirus (NP_604475.1), Rhizoctonia solani dsRNA virus 2 (YP_009011230.1), White clover cryptic virus 1 (YP_086754.1), Chondrostereum purpureum cryptic virus 1 (CAQ53729.1), Flammulina velutipes browning virus (BAH56481.1), Helicobasidium mompa dsRNA mycovirus (BAC23065.1), Rhizoctonia solani dsRNA virus 3 (YP_009329886.2), Heterobasidion partitivirus 12 (AHL25151.1), Trichomonas vaginalis virus 1(AAA62868.1).

**Figure 4 viruses-13-02254-f004:**
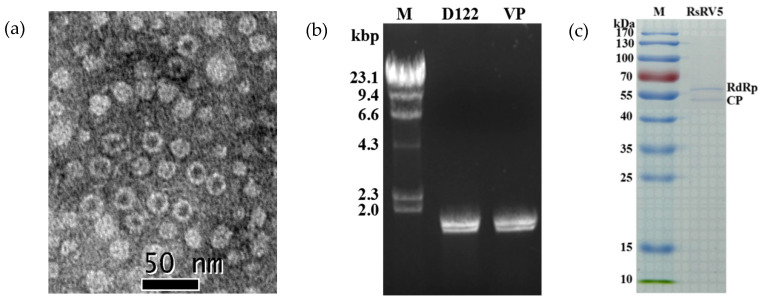
Viral particle traits in strain D122 of *Rhizoctonai solani* AG-1 IA. (**a**) Viral particles observed under TEM (negative staining). Particles were purified from mycelia of strain D122 (scale bars, 50 nm); (**b**) Agarose gel electrophoresis of dsRNAs (dsRNA-1 and dsRNA-2) extracted from mycelia of strain D122 and from viral particles (VP), respectively. M: molecular markers (λ DNA digested with *Hind* III); (**c**) SDS-PAGE analysis of structural proteins from viral particles. The size of the Coomassie blue-stained protein was estimated by comparison with protein markers. kDa: Kilodaltons.

**Figure 5 viruses-13-02254-f005:**
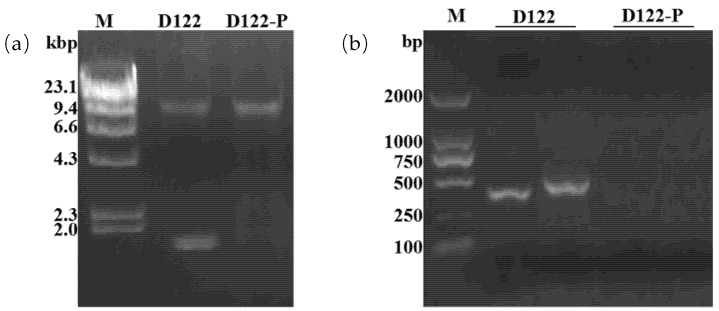
Detection of RsRV5 in strains D122 and D122-P of *Rhizoctonai solani* AG-1 IA. (**a**) Detection of dsRNA of RsRV5 from a derived strain obtained from protoplast regeneration method. M, λDNA Hind III markers, D122 is the original mycovirus-infected strain, D122-P is the derived strain obtained from protoplast regeneration method; gDNA: genomic DNA of the fungal strain D122. (**b**) RT-PCR detection of RsRV5 from isogenic strains D122 and D122-P.

**Figure 6 viruses-13-02254-f006:**
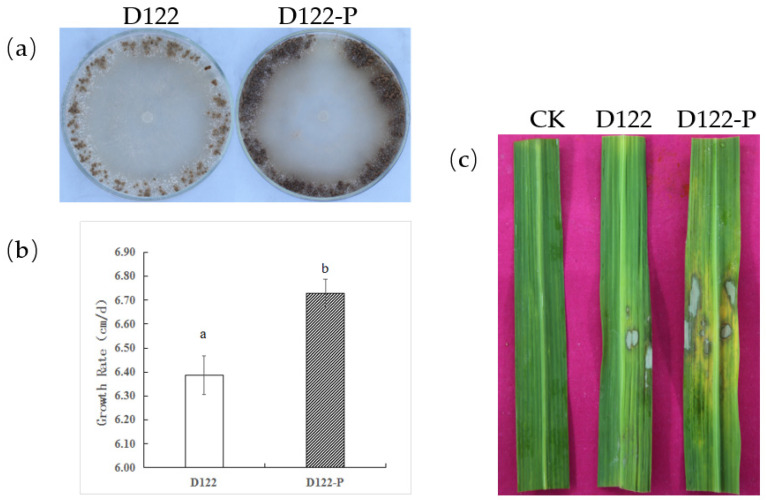
Hypovirulence-associated traits in strain D122 of *Rhizoctonai solani* AG-1 IA. (**a**) Colony morphology of strains D122 and D122-P after 4 days of culture on PDA in the dark; (**b**) comparison of average mycelial growth rate on PDA plates of the strains D122 and D122-P. The lowercase letters (a and b) on top of the bars in b indicate whether the differences are statistically significant (*p* < 0.05); (**c**) Pathogenicity. The symptoms on detached rice leaves caused by strains D122 and D122-P at 28 °C for 72 h.

**Figure 7 viruses-13-02254-f007:**
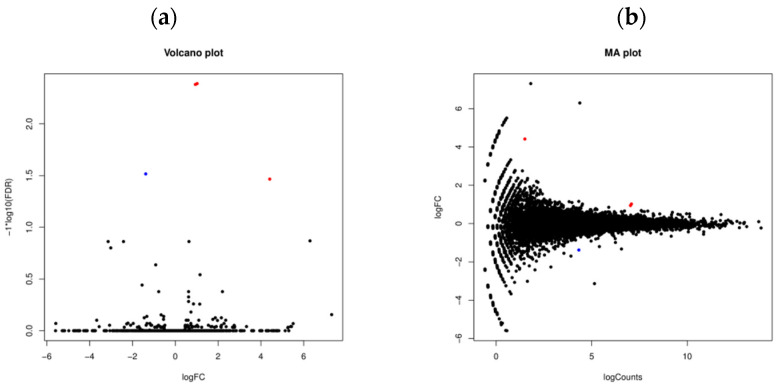
Differentially expressed genes between two isogenic strains, D122 and D122-P of *Rhizoctonia solani* AG-1 IA, by using the transcriptomic technique. (**a**) Volcano plot displaying -log10 (FDR) on the Y-axis, indicating that a larger value is a more significant difference, and logFC on the X-axis, indicating the extent of the difference in gene expression between the two samples; (**b**) MA plot displaying logCounts on the Y-axis, indicating that the higher the value, the higher the expression of the gene, and logFC on the X-axis. Red and blue represent up-regulated and down-regulated genes, respectively, while black represents genes that are not significantly differentially expressed.

**Figure 8 viruses-13-02254-f008:**
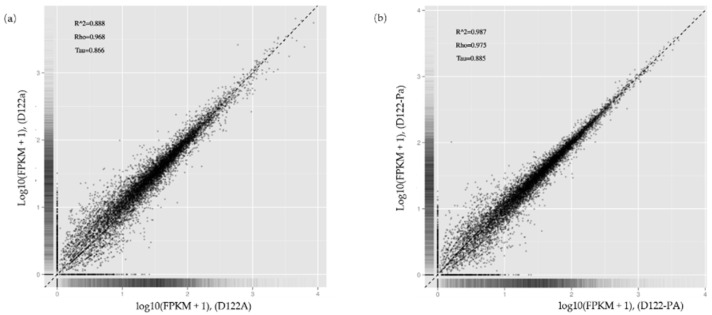
Correlation check of RNA-seq. (**a**) Correlation check of samples D122a and D122A; (**b**) Correlation check of samples D122-Pa and D122-PA. R^2 is the square of the Pearson Correlation Coefficient, Rho is the Spearman Correlation Coefficient and Tau is the kendall-tau Correlation Coefficient.

**Table 1 viruses-13-02254-t001:** Statistics of alternative splicing of RNA-seq samples.

Alternative Splicing Types	Alternative Splicing Numbers	Numbers of Genes Involved
exon skipping	7792	2662
intron retention	13,377	3965
alternative 5′ss splice or alternative 3′ss splice	18,396	5088
alternative transcription start site	13,466	10,687
alternative transcription terminal site	12,504	10,687

## Data Availability

The sequences reported in the present manuscript have been deposited in GenBank with accession numbers MG597242 and MG597241.

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
