# Peer review of "Genome Organizations and Functional Analyses of a Novel Gammapartitivirus from Rhizoctonia solani AG-1 IA Strain D122"

_viruses, 2021, doi:10.3390/v13112254_

Round 1

Reviewer 1 Report

The authors described a mycovirus designated Rhizoctonia solani dsRNA virus 5 (RsRV5) from strain D122 of Rhizoctonia solani AG-1 IA, the causal agent of rice sheath blight. Sequence alignments and phylogenetic analyses showed that RsRV5 is a new member of the genus Gammapartitivirus in the family Partitiviridae. RsRV5-cured strain D122-P possessed the traits of recovered mycelial growth, increased pigmentation sclerotia production and hypovirulence to rice leaves compared with wild type RsRV5-infection strain D122. Transcriptome analysis showed that 3 genes were differentially expressed between two isogenic strains D122 and D122-P. These findings provided new insights into the molecular mechanism of the interaction between RsRV5 and its host D122 of R. solani AG1-IA.

    I have several comments to improve the manuscript.

    Please follow them carefully.

Major comments:

  1. L17. The authors should show the buoyant density of isometric virus particles in CsCl.
  2. L20. Possessed the traits of “recovered” mycelial growth.
  3. L31. It is not correct. The numbers of ssRNA mycoviruses are much more rather than dsRNA mycoviruses.
  4. L112-114. This procedure is not accurate. Please describe protocol for northern hybridization. How to denature the RNA molecules on the membrane or on the agarose gel?
  5. L135. Here, you cite two references, but only one author's name, Hillman et al. The authors should add Sasaki et al.
  6. L136. There is no detail description for the procedure of ribavirin treatment. Please describe the procedures and processes to get virus-free strain D122.
  7. L191. I think the authors had wrong idea. The ORF1 started at the nt 30 and ended nt 1787.
  8. L248. According to the sequence data of dsRNA1, the ORF1 predicts to start at the nt 30 and end nt 1787. Therefore, the authors should investigate the N-terminal amino acid sequences, or peptides finger mass finger printing analysis for the RdRp band of RsRV5. If possible, investigate the amino acid sequences of the CP protein also.
  9. L267: The authors should describe the procedures to isolate virus-free strains. How many candidate mycelia did the authors screening for virus-free strains?
  10. L321-335. These sentences should move to Discussion.
  11. L356. What kinds of responsibilities do the authors expect? Please explain the responsibilities for virus replication cycle which are affected by the interrupted poly A tails.
  12. L360-361: There is no evidence. The authors should show the facts to prove the separate encapsidation of the dsRNA segments using by CsCl density gradients methods.
  13. L389-393. Please add the final conclusion here. Do not end with other reports’ results.

Minor points.

  1. References: 25. Atsuko, S.; Satoko, K.; Mari, O.; Yuri, O.; Hitoshi, N.; Kouji, Y. Artificial infection of Rosellinia necatrix with purified viral particles of a member of the genus Mycoreovirus reveals its uneven distribution in single colonies. Phytopathology 2007, 97, 278-286.

Reviewer 2 Report

The MS describes genome organization and some functional consequences of a novel gammapartitivirus from Rhizoctonia solani strain. 

This paper is good organized and clearly written, with a few reservations: 

In Introduction, according to most papers, the genome of CHV1 hypovirus is dsRNA, but not ssRNA. 

The number of novel partitiviruses increasing rapidly, it should be worthy to mention that outside the approved genera, epsilon- and zetapartitivirus genera are proposed (https://doi.org/10.1371/journal.pone.0219207) also, or note amount of unclassified viruses here.

In Material and Methods, correct expression on lines 102-103, correct typos on line 113, and provide orientation of specific primers (5´, and 3´ends). Provide more details (parameters, explanation of FC, and FDR values, respectively) about the computational analysis  of differently expressed genes, here.

There is mistake in Results (line 192), where the ORF is ending at nt 1787, but not at 1235! There are missing sequence details of almost all gammapartitiviruses mentioned on Figure 3, on the other hand there are Discula destructiva virus 1, and Ophiostoma partitivirus 1 data, which are not present in the figure.

Finally, Discussion is not real discussion, but repetition of results/abstract. More thorough comparison of RsRV5 with the corresponding related viruses should be performed as well as the hypovirus phenomenon should be evaluated/compared/discussed here. If you have decided to use the abbreviation of journal with dots, please, strictly follow it with all references. In ref. 8 and 12, U.S.A. is missing.

Correct also the typography of the AG-1 IA strain, which is used in several styles (AG1-IA, AG-1IA, AG-1 IA). 

Round 2

Reviewer 1 Report

I hope that the authors willl show the buoyant density of isometric virus particles in CsCl for the next paper.

I also hope that the authors should investigate the N-terminal amino acid sequences, or peptides finger mass finger printing analysis for the RdRp band of RsRV5. If possible, investigate the amino acid sequences of the CP protein for the next paper.